# Transcriptomic analysis to elucidate the response of honeybees (Hymenoptera: Apidae) to amitraz treatment

**Liang Ye[‡], Peng Liu[‡], Tengfei Shi[‡], Anran Wang, Yujie Zhu, Lai Li, Linsheng Yu***

Anhui Agricultural University, Hefei, China

‡ Co-first author
* yulinshengahau@163.com

**Data Availability Statement:** All relevant data are within the paper and SRA database(https://dataview.ncbi.nlm.nih.gov/object/PRJNA593612?reviewer=elfpv4vmb047ik7k9efhmla825).

## Abstract

Amitraz is an acaricide that is widely used in apiculture. Several studies have reported that in honeybees (*Apis mellifera* Linnaeus; Hymenoptera: Apidae), amitraz affects learning, memory, behavior, immunity, and various other physiological processes. Despite this, few studies have explored the molecular mechanisms underlying the action of amitraz on honeybees. Here, we investigated the transcriptome of honeybees after exposure to 9.4 mg/L amitraz for 10 d, a subchronic dose. Overall, 279 differentially expressed genes (DEGs) were identified (237 upregulated, 42 downregulated). Several, including *Pla2*, *LOC725381*, *LOC413324*, *LOC724386*, *LOC100577456*, *LOC551785*, and *P4504c3*, were validated by quantitative PCR. According to gene ontology, DEGs were mainly involved in metabolism, biosynthesis, and translation. Kyoto Encyclopedia of Genes and Genomes pathway analyses revealed that amitraz treatment affected the relaxin signaling pathway, platelet activation, and protein digestion and absorption.

## Introduction

Approximately 80% of flowering plants, including many crops, require insects to pollinate [1–3]. Honeybees (*Apis mellifera* Linnaeus; Hymenoptera: Apidae) are the most important pollinators worldwide [4]. One-third of global food is linked to the pollination activity of honeybees [4]. In recent years, the substantial decline in apiculture has garnered much attention [5–8]; however, the underlying reasons for this remain poorly understood. Many factors affect the wellbeing of honeybees, including pathogens, pesticides, malnutrition and changing apicultural practices [9–11]. Among these, pesticide exposure has been widely accepted to be the major contributor to a decline in the honeybee population [10].

Honeybees are continuously exposed to agricultural pesticides, which are transported to hives by foraging bees [10]. Direct application of acaricides within beehives to control *Varroa* mites and other pests creates an extra pesticide burden on the bees [10]. Acaricides and pesticides contaminate apicultural products such as honey, beeswax, and pollen [8, 12, 13].

The midgut of honeybee is an absorptive organ and involved in degraded chemical compounds[14]. The epithelium, in particular, is responsible for detoxification of ingested

**Funding:** This work was supported by the Earmarked Fund for China Agriculture Research System (No. CARS-45-KXJ10). The funder had no role in study design, data collection and analysis, decision to publish, or preparation of the manuscript.

**Competing interests:** The authors have declared that no competing interests exist.

xenobiotics[15]. Meanwhile, honey bee larvae exposed to sublethal concentrations of a broad range of pesticides resulted in midgut cell apoptosis[16]. As the same time, the midgut is the principal barrier to invasion of the honey bee for many pathogens [17].

Amitraz [1,5-di-(2,4-dimethylphenyl)-3-methyl-1,3,5-triaza-penta-1,4-diene] is a formamidine pesticide used globally to control pests on animals and crops [18]. It is an acaricide and mainly acts on the central nervous system of ectoparasites by interacting with octopamine receptors, causing lethal and sublethal effects [19]. In the apiary, beekeepers can control *Varroa* mites by fumigation of beehives with amitraz, but it results in contamination of honey stored in combs[20]. Amitraz does not persist in the hive environment [21], but its metabolite N-(2,4-dimethylphenyl)-N′-methylformamidine can accumulate and has been found in wax, pollen, and inside the bees themselves [13]. Moreover, acute exposure to amitraz also can kill honeybee larval midgut epithelial cells [22]. Amitraz also affects learning, memory [23, 24], immunity [25] and sensory organs [26] in honeybees. In addition, a recent study reported that amitraz affected the immune system of the queen [27]. Amitraz stress leads to increase glutathione S-transferases activity in the larval instars, pupae, newly emerged bees and nurse bees [28].

Despite the adverse effects of amitraz on honeybees, the relevant molecular mechanisms remain poorly explored. In this study, we conducted high-throughput RNA sequencing (RNA-Seq) analyses to investigate honeybee transcriptomes after exposure to 9.4 mg/L amitraz for 10 d, a subchronic level. Differentially expressed genes (DEGs) were identified and analyzed. Our aim was to help understand the molecular mechanisms underlying the action of amitraz to elucidate reasons for the decline in honeybee populations.

## Materials and methods

### Honeybee rearing

The honeybees were obtained as previously described [29, 30]. Two frames with sealed broods nearing adult emergence were collected from an apparently healthy colony at the Institute of Apiculture Research, Anhui Agriculture University, Hefei, China. The population had not been exposed to pesticides. The frames were held in darkness at 35 ± 1°C with relative humidity (RH) 50% ± 10%. Newly emerged honeybees were then placed into wooden cages (11 × 10 × 8 cm) in darkness for 2 d (28 ± 1°C, RH 60% ± 10%). Throughout the experimental period, bees were fed sufficient fresh pollen and 50% (w/v) sucrose-water solution. Dead bees were removed daily.

### Amitraz treatment

We followed previously reported methods by Shi et al. with some minor modifications[31]. The median lethal concentration ($LC_{50}$) of amitraz to honeybees is 94 mg/L [32]. Herein we used amitraz(99% purity) which was obtained from aladdin company(Shanghai, China) at a sublethal concentration (9.4 mg/L). A stock solution of amitraz (1000 mg/L) was prepared in acetone. Working solution (9.4 mg/L) was prepared by dissolving the stock solution in 50% sucrose-water solution. Sucrose–water solution added the equivalent acetone without amitraz as a negative controls.

Three-day-old bees were used for assays (45 bees/replicate, three replicates/treatment). After 10 d, all bees were collected and placed at 4°C for 5 min to anesthetized them; the bees were then dissected on ice, using liquid nitrogen to flash freeze the sample and the midgut was removed and stored at −80°C.

## RNA extraction, library preparation and sequencing

Ten midguts from each replicate were pooled for RNA extraction using TRIzol reagent (Invitrogen, Carlsbad, CA, USA). RNA concentration was quantified and RNA integrity verified. Sequencing libraries were generated using a NEBNext Ultra RNA Library Prep Kit for Illumina (NEB, CA, USA) following the recommended protocol; 3 μg RNA from each sample were used to prepare the library. Index codes were added to link sequences with the sample from which they originated. mRNA was purified from total RNA using poly-T oligo-attached magnetic beads. mRNA was fragmented in 5× NEBNext First Strand Synthesis Reaction Buffer at elevated temperature. First strand cDNA was synthesized using random hexamer primers and M-MuLV Reverse Transcriptase (RNase H). Second strand cDNA was synthesized using DNA Polymerase I and RNase H. Overhangs were blunted using exonuclease/polymerase. 3′-ends of DNA fragments were adenylated and ligated with NEBNext Adaptors. DNA fragments (150–200 bp long) were selected by purification using an AMPure XP system (Beckman Coulter, Beverly, MA, USA). USER Enzyme (3 μL; NEB) was incubated with the size-selected, adaptor-ligated cDNA at 37°C for 15 min, then 5 min at 95°C. PCR was performed with Phusion High-Fidelity DNA polymerase, universal PCR primers. The amplicons obtained were purified using the AMPure XP system, and library quality was assessed on an Agilent Bioanalyzer 2100. Clustering of the index-coded samples was performed on a cBot Cluster Generation System using the TruSeq PE Cluster Kit v3-cBot-HS (Illumina). Then, the library preparations were sequenced (Illumina HiSeq 4000); 150-bp paired-end reads were generated.

## Read processing

Raw reads (FASTQ) were initially processed using in-house Perl scripts. Clean reads were obtained by removing low-quality reads, and those containing adapter sequences or poly-N. All downstream analyses used high-quality clean reads. At the same time, the Q20, Q30, and GC contents were calculated. The index of the honeybee genome (NCBI: assembly Amel_-HAv3.1) was built using Bowtie v2.2.3, and reads were aligned to the genome using TopHat v2.0.12. HTSeq v0.6.1 was used to count read numbers mapped to each gene.

## Analysis of differential expression

Differential expression analysis used the DESeq R package v2.15.3. The resulting $P$-values were adjusted using the Benjamini and Hochberg approach for controlling false discovery rate; genes with adjusted $P$-value <0.05 were considered to be differentially expressed. Gene ontology (GO) enrichment analysis of DEGs was performed using the GOseq R package (gene length bias was corrected); GO terms with corrected $P$-values <0.05 were considered significantly enriched by DEGs. For Kyoto Encyclopedia of Genes and Genomes (KEGG) pathway analysis, we used KOBAS to assess the statistical enrichment of DEGs.

## Real-time quantitative PCR (qPCR) analysis

We selected seven DEGs identified following RNA-Seq (*LOC725381*, *CYP4C3*, *LOC41332*, *Pla2*, *LOC724386*, *LOC100577456*, and *LOC551385*) for verification by qPCR analysis. *RpS5* and *β-actin* was used as the reference gene, and all the primers used were given in Table 1. We used 0.5 μg of total RNA (same as the total RNA for RNA-seq) for each sample. ReverTra Ace qPCR RT Master Mix Kit (Toyobo, Osaka, Japan) and a SuperReal PreMix Plus (SYBR Green) Kit (TIANGEN, Beijing, China) were used to obtain cDNA and perform qPCR, respectively. The relative expression levels of genes was calculated using the $2^{-\Delta\Delta Ct}$ method [33]. Primer sets are listed in Table 1. Student's *t*-test was used to assess differences in gene expression levels

**Table 1. Primer sequences.**

| Genes | Primer Sequences (5'~3') | amplification efficiency (%) |
|---|---|---|
| LOC725381 | Forward: CTAACCGCATTTCCCTTT | 95.6 |
| | Reverse: ATTCCGCATACAACAACG | |
| CYP4C3 | Forward: ATTTGTCTTGCGATGAGC | 97.8 |
| | Reverse: ACGACGAAACAGTAGGGA | |
| LOC413324 | Forward: ATTGGCGGCACTCCTGAT | 101.5 |
| | Reverse: TCCACGGGAAGGCGATTA | |
| Pla2 | Forward: GCGACGACAAGTTCTATGAT | 95.9 |
| | Reverse: GTAGTGAAGACAACGACCCTC | |
| LOC724386 | Forward: CATTTTGTTCTGGGAGTGGGT | 96.8 |
| | Reverse: CGTATTTGCGGTGCTCTTCAT | |
| LOC1100577456 | Forward: CGTTCTCCTCGCTTATACCGT | 100.4 |
| | Reverse: GAATGATTTCAGCCCTCCACT | |
| LOC551385 | Forward: CTTGCTGCCCTCCCGAAACTC | 103.7 |
| | Reverse: CGAGAACACGCCGCAGAAAAG | |
| RpS5[34] | Forward: AATTATTTGGTCGCTGGAATTG | 99.5 |
| | Reverse: TAACGTCCAGCAGAATGTGGTA | |
| β-actin[35] | Forward: TGCCAACACTGTCCTTTCTG | 95.2 |
| | Reverse: AGAATTGACCCACCAATCCA | |

between amitraz-treated bees and controls. We used three replicates per group for qPCR validation.

## Results

### Survival

As shown in Fig 1, the average survival rate on day ten for the forager bees exposed to 9.4 mg/L amitraz and control of 50% sucrose-water solution were 77.8% and 88.9%, respectively. There were no significant differences among all treatments (Log-rank $\chi^2$ = 2.024, df = 1, P = 0.1548; Fig 1). The 9.4 mg/L amiteaz is a sublethal concentration.

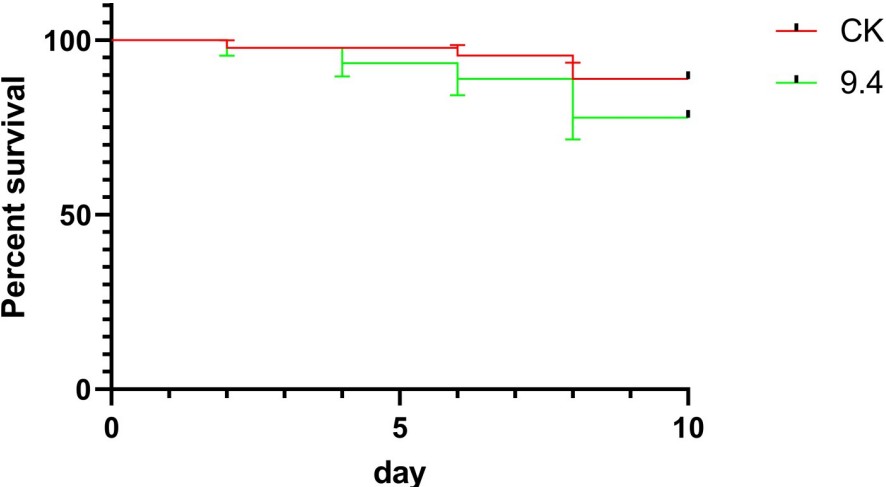

**Fig 1. Survival of forager bees subjected to chronic exposure to field-realistic concentrations of amitraz after 10 d.**

## Raw read processing and quantitative gene expression

Six libraries were created from amitraz-treated bees and controls: amitraz-1, amitraz-2, amitraz-3, control-1, control-2, and control-3, which generated 46588584, 38742446, 39577344, 42787014, 47048748, and 44719946 usable reads, respectively. The Q20 were 99.97%, 99.97%, 99.97%, 99.97%, 99.97% and 99.97%, respectively, while the Q30 were 97.04%, 97.06%, 96.99%, 97.00%, 97.30% and 96.93%, respectively. Q20, Q30, and GC contents were listed in S2 Table. After mapping to the reference genome (NCBI: assembly Amel_HAv3.1) and junction database, 44944558, 37354986, 38094470, 41341014, 45468272 and 43453530 total mapped reads were acquired, respectively. The numbers of uniquely mapped reads were 27182392, 21407197, 24815509, 26929339, 28752629 and 27107676 respectively. Among these unique reads, 57.31%–65.14% were mapped to exons (S1 Table). The sequencing data are available in the SRA database (https://dataview.ncbi.nlm.nih.gov/object/PRJNA593612?reviewer=elfpv4vmb047ik7k9efhmla825) of the NCBI system.

The average number of genes expressed in the treatment and control groups was 11410 and 11303, respectively; 11034 genes were expressed in both groups (Fig 2).

In each library, 9.91%–11.74% of reads had reads per kilobase per million mapped reads (RPKM) values of <0.1; 9.76%–10.98% of reads had RPKM values of 0.1–0.3; 36.58%–37.69% of reads had RPKM values of 0.3–3.57; 19.12%–20.62% of reads had RPKM values of 3.57–15; 11.25%–12.29% of reads had RPKM values of 15–60; and 9.41%–10.07% of reads had RPKM values of >60 (S3 Table). Thus, a few genes were expressed at very high levels, but most were expressed at low levels, indicating that the distribution of our gene expression dataset was normal.

## DEGs, GO enrichment analysis, and KEGG pathway analysis

Overall, 279 DEGs were detected in honeybees exposed to 9.4 mg/L amitraz for 10 d: 237 (84.9%) were upregulated and 42 (15.1%) were downregulated (Fig 3 and S4 Table). S5 Table lists the 23 most significantly differentially expressed genes; of these, one was downregulated and 22 were upregulated. Fig 4 shows the 30 most enriched GO terms. In GO classification "biological process", most DEGs were involved in translation and metabolic and biosynthetic processes. In category "cellular components", most DEGs were associated with the ribosome. Finally, considering classification "molecular function", most DEGs were enriched in structural constituents of the ribosome, structural molecule activity, and oxidoreductase activity.

In total, 135 DEGs (116 upregulated and 19 downregulated) were mapped to 67 KEGG pathways, 10 of which were significantly enriched (Table 2).

## qPCR analysis

To validate our RNA-Seq data, seven DEGs (*LOC725381*, *CYP4C3*, *LOC41332*, *Pla2*, *LOC724386*, *LOC100577456*, and *LOC551385*) were checked by qPCR. Consistent with our sequencing data, *LOC724386* was downregulated in amitraz-treated bees, while *LOC725381*, *CYP4C3*, *LOC41332*, *Pla2*, *LOC100577456*, and *LOC551385* were upregulated (Fig 5) (The reference gene is *RPS5* and *β-actin*, *LOC725381*: $t = 18.978$, df = 2, $P = 0.0276$; *CYP4C3*: $t = 3.165$, df = 2, $P = 0.0258$; *LOC413332*: $t = 15.623$, df = 2, $P = 0.0262$; *Pla2*: $t = 15.812$, df = 2, $P = 0.0063$; *LOC724386*: $t = -19.756$, df = 2, $P = 0.0095$; *LOC100577456*: $t = 8.232$, df = 2, $P = 0.0092$; *LOC551385*: $t = 4.551$, df = 2, $P = 0.0226$).

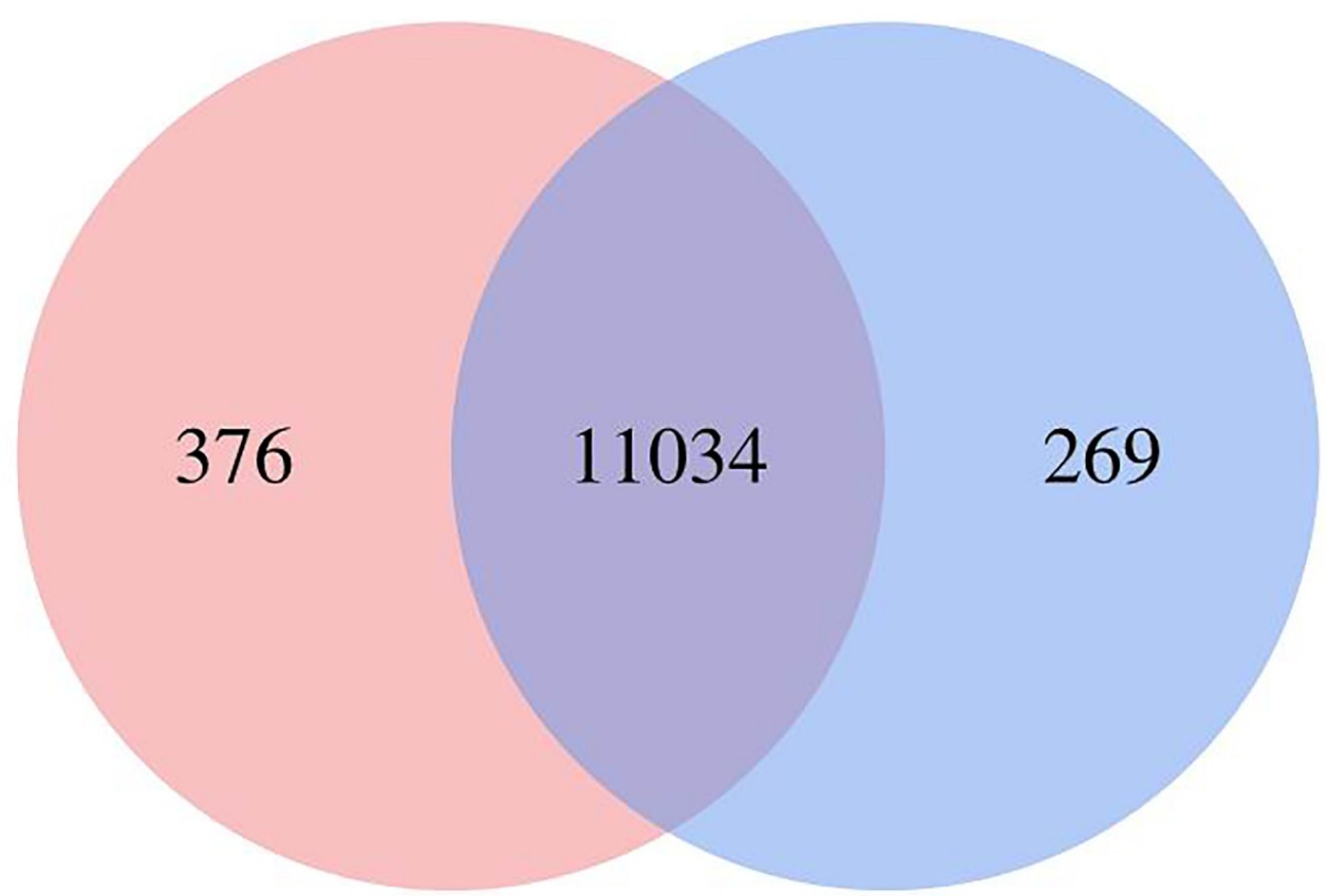

**Fig 2. Average number of genes specifically expressed in amitraz and control libraries.** Shown as the number of genes expressed in each class.

## Discussion

Herein, we exposed honeybees to 9.4 mg/L amitraz for 10 d, which led to the identification of 279 DEGs (237 upregulated and 42 downregulated genes) in the honeybee transcriptome. In order to further study the metabolic pathways influenced in honey bees after amitraz exposure, 91 detailed related pathways of the differential genes were constructed using KEGG pathway analysis. Among these, four pathways, relaxin signaling pathway, platelet activation, protein digestion and AGE-RAGE signaling pathway in diabetic complications were extremely significantly affected (P<0.01). Previous reviews have showed that a relaxin was a factor communicating abnormal growth status of Drosophila larval imaginal discs to the neuroendocrine centers that control the timing of the onset of metamorphosis.[36, 37]. Herein, we found that the relaxin signaling pathway were activated in honeybees after exposure to amitraz, which indicates that amitraz potentially influenced developmental processes of honeybees. To facilitate feeding, certain hematophagous invertebrates possess inhibitors of collagen-induced

## AmitrazVSControl

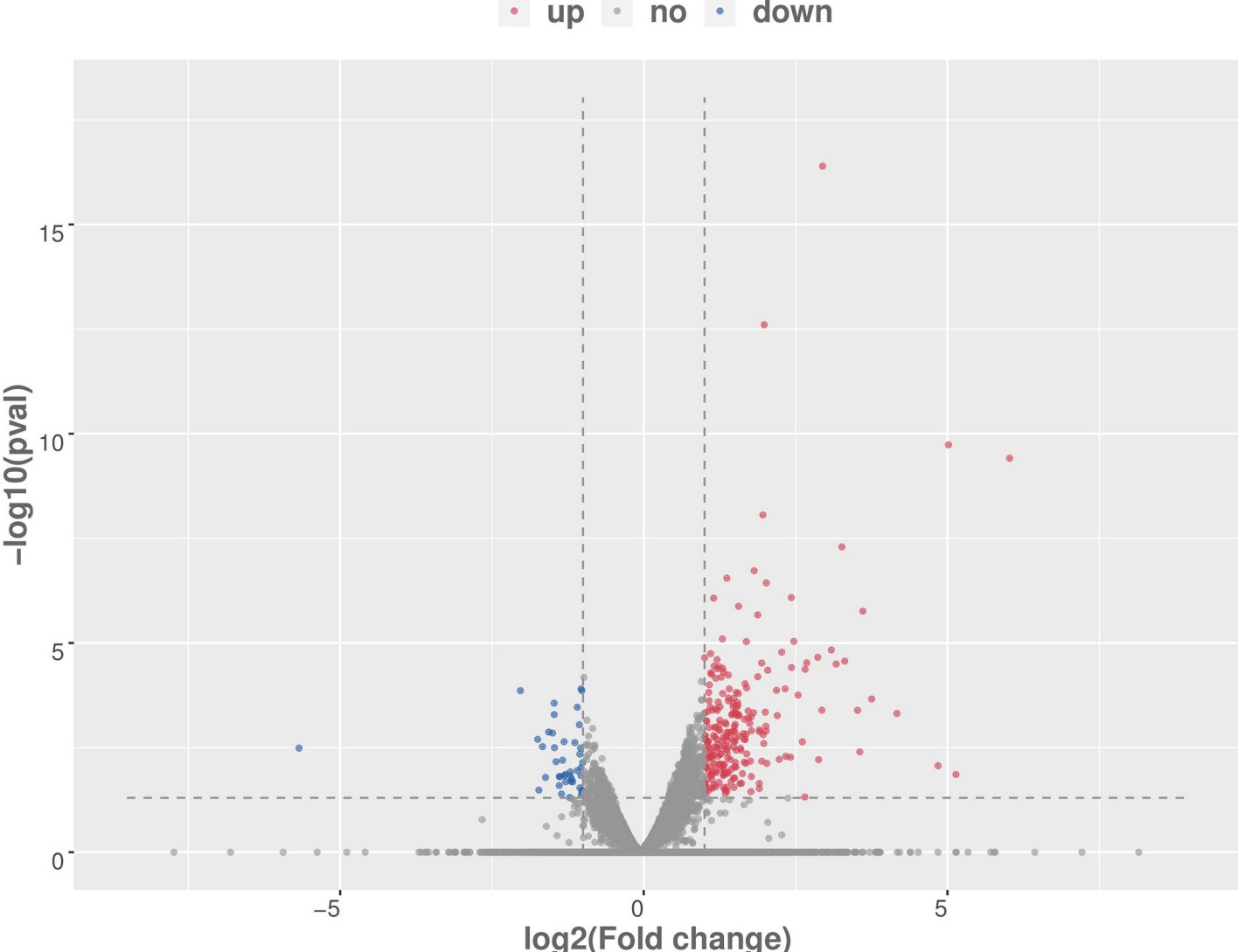

**Fig 3. Volcano plot of differentially expressed genes in honeybees exposed to 9.4 mg/L of amitraz for 10 days.** Genes with an adjusted *P* value of <0.05 (FDR correction method) were considered to be differentially expressed. Red: upregulated genes in amitraz-treated bees; green: downregulated genes in amitraz-treated bees; blue: no significant difference.

platelet aggregation in their saliva, inhibited platelet aggregation need inhibit signal transduction necessary for platelet activation by collagen[38]. The hemocytes phagocytosis may play an important role in the cellular immune responses in insects, and the platelet-activating factor can influence phagocytosis of cells[39]. In our study, up-regulation of two genes (*LOC113219 380* and *LOC113219382*) of Platelet activation pathway after exposure to amitraz (*P* <0.01), indicating that amitraz might influence the honeybees immunity. Previous study indicated that imidacloprid was involved in the intoxication of honeybees, it could compromise the viability of the midgut epithelium and affected protein digestion and absorption[40]. Comparison of transcriptome profiling between HearNPV-infected and control healthy *Helicoverpa armigera* larvae during an early stage post-inoculation, KEGG analysis indicated an enrichment of

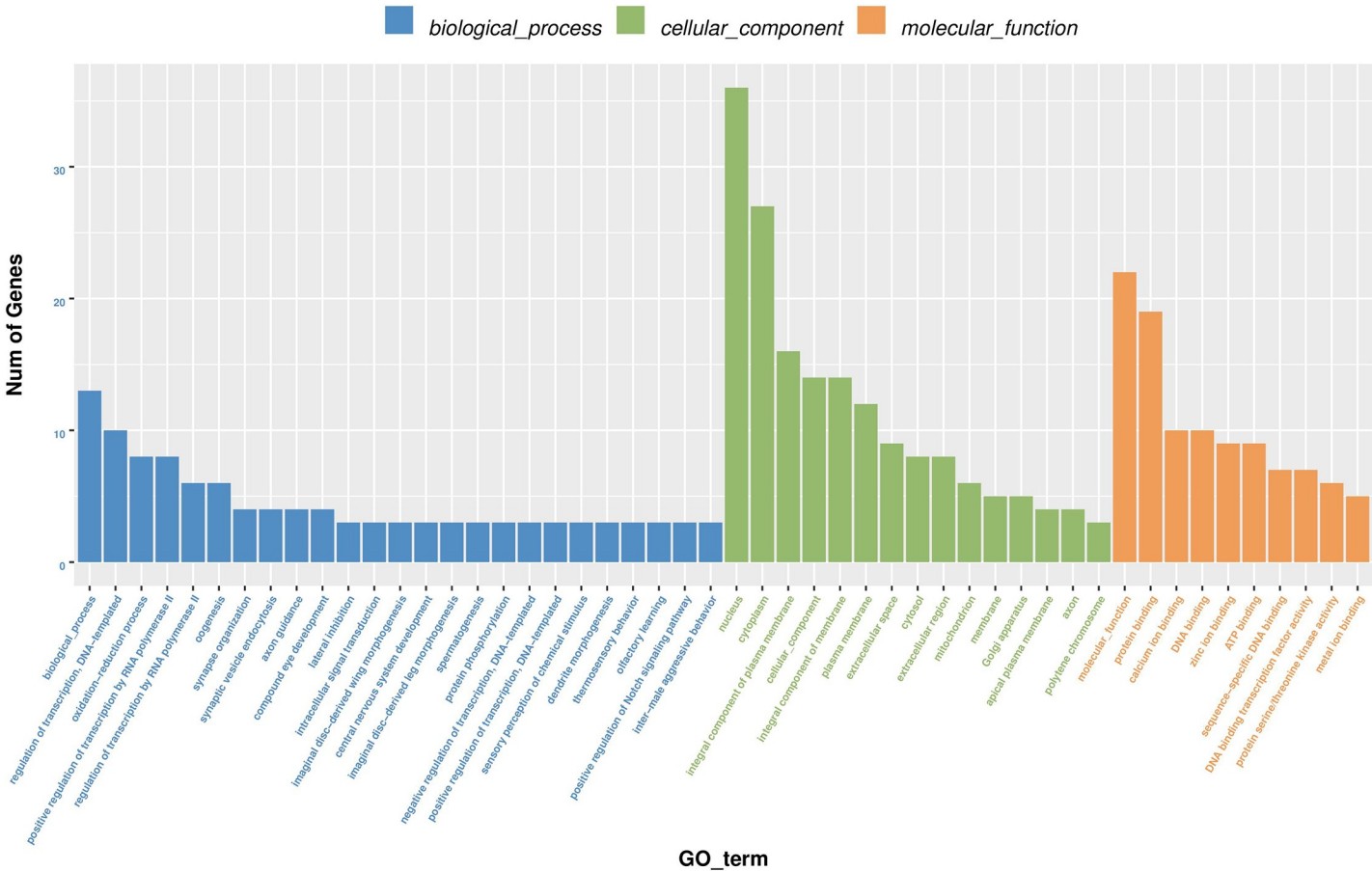

**Fig 4. Gene ontology enrichment analysis of differentially expressed genes (DEGs) in honeybees exposed to 9.4mg/L amitraz for 10 days.** Green bars: DEGs enriched for biological process; orange bars: DEGs enriched for cellular components; purple bars: DEGs enriched for molecular function. * indicates that GO terms were significantly enriched by DEGs (corrected *P* values of <0.05, FDR correction method).

these differently expressed genes some pathways, including protein digestion and absorption, proved that the DEGs participated in nutritional digestion and exhibited specific expression patterns in a continuous time-course assessment[41]. In this study, we believe that in response to amitraz challenge, honeybees could repair the damages by inducing the expression levels of

**Table 2. The five significantly enriched pathways, corrected *P*-value < 0.05.**

| Pathways | Pathway ID | Genes number | Corrected *P*-value |
|---|---|---|---|
| Relaxin signaling pathway | ko04926 | 2 | 0.01 |
| Platelet activation | ko04611 | 2 | 0.01 |
| Protein digestion and absorption | ko04974 | 2 | 0.01 |
| AGE-RAGE signaling pathway in diabetic complications | ko04933 | 4 | 0.01 |
| Amoebiasis | ko05146 | 2 | 0.01 |
| Cutin, suberine and wax biosynthesis | ko00073 | 2 | 0.02 |
| Cellular senescence | ko04218 | 2 | 0.03 |
| Leukocyte transendothelial migration | ko04670 | 1 | 0.04 |
| Taste transduction | ko04742 | 1 | 0.04 |
| D-Glutamine and D-glutamate metabolism | ko00471 | 1 | 0.04 |

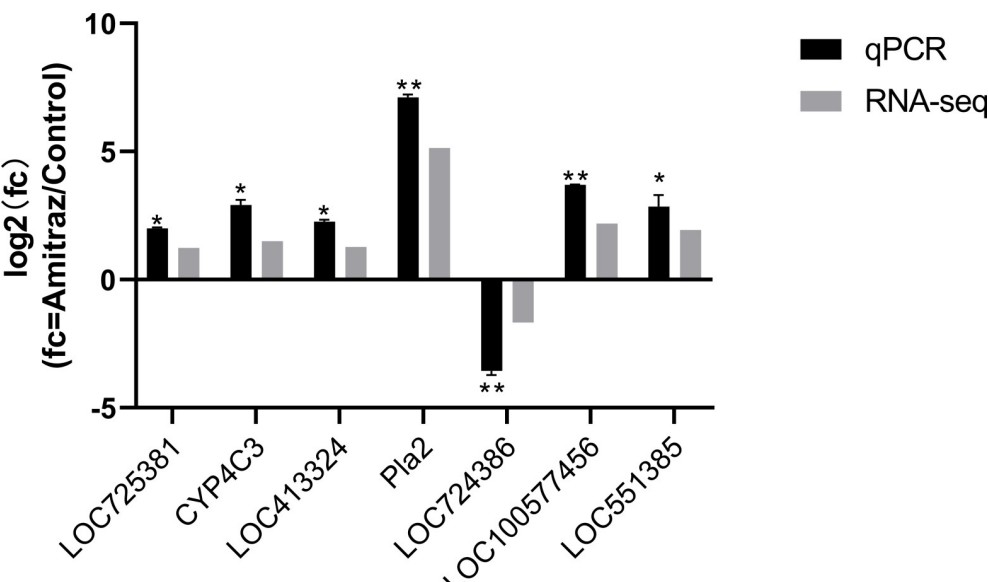

**Fig 5. Real-time quantitative PCR and RNA-seq analysis of *LOC725381*, *CYP4C3*, *LOC41332*, *Pla2*, *LOC724386*, *LOC100577456*, and *LOC551385* genes expression multiple in honeybees exposed to 9.4 mg/L of amitraz for 10 days.** Reference gene were *RPS5* and *β-actin*. Values represent means ± SEM. * indicates a significant difference in comparison with controls ($P < 0.05$) and ** indicates a statistically significant difference in comparison with controls ($P < 0.01$).

the protein digestion and absorption pathway. The study showed that six key pathways might be associated with longevity of *Drosophila* including the AGE-RAGE-signalling pathway in diabetic complications[42]. Comparison between control and Cr (VI)-treated samples of mantis shrimp, AGE-RAGE signaling pathway in diabetic complications were significantly enriched[43]. In honeybees, AGE-RAGE signaling pathway in diabetic complications governs the neural activity to drive the age-specific labor division[44]. In this study, expression levels of four genes (*LOC113219380*, *LOC113219382*, *Plc* and *LOC724607*) of AGE-RAGE signaling pathway in diabetic complications upregulated by exposure to amitraz ($P <0.01$), Chouquet et al. clarified the role of *Plc* in *Spodoptera littoralis* olfactory transduction[45]. Our study suggest that amitraz probably affected longevity, developmental processes and olfactory transduction of honeybees.

At the individual level, honeybees elicit both cellular and humoral innate immune responses against extraneous substances [19]. Antimicrobial peptides (AMPs) are a class of peptides with low molecular weight; they are encoded by specific genes and are important effectors of natural immunity [46]. Many studies have reported that pesticides affect immunocompetence by regulating the gene expression levels of AMPs in honeybees. For example, exposure to imidacloprid caused most immune related AMP genes (encoding *apidaecin*, *hymenoptaecin* and *defensin-1*) to be downregulated in white- and brown-eyed pupae, but in adults caused an increase in honey bee immune response[47]. Thiamethoxam treated honey bees were further exposed to either thiamethoxam or *Nosema*, which caused AMP genes *abaecin*, *defensin-1* and *defensin-2* to be upregulate[48]. In this study, AMP genes, like apidaecin (*Apid1*), were induced in amitraz-treated bees, indicating that amitraz also triggers the immune response in honeybees.

Among the identified DEGs, four serine/threonine-protein kinase (STK) genes were present: *STK CG31145*, *STK A2*, *STK PAKm*, and *STK MKNK1II*. STKs are enzymes involved in metabolism, cell differentiation, gene expression, disease resistance, and other processes [49,

50]. STKs participate in stress resistance in insects [51]. In honeybees, STKs are related to cold and heat stress [52, 53]. Herein, we found that the expression levels of *STK CG31145*, *STK A2*, *STK PAKm*, and *STK MKNK1II* were upregulated by exposure to amitraz, which indicates that they are potentially involved in conferring tolerance to amitraz.

Further, we found that two detoxification-related genes, *CYP4C3* (encoding cytochrome P450 4c3) and *CaE-I1* [encoding carboxylesterase (CarE) clade I], were differentially expressed. Cytochrome P450 monooxygenase (CYP) enzymes have been linked to insecticide resistance (i.e., detoxification) or environmental response [54]. Many xenobiotics and pesticides are metabolized by CYPs, such as pyrethroid lambda-cyhalothrin [55, 56], neonicotinoid insecticides [57, 58], aflatoxins [59], and the organophosphate coumaphos [60]. Honey bee are known to have the far fewer numbers of CYP family genes compared to other insects[61, 62], and some subfamilies that have been analyzed include CYP6 and CYP9. The CYP6 subfamily is insect-specific [54] and is involved in phytochemical metabolism [63]. The CYP9 subfamily is responsible for the degradation of various classes of plant protection products [56, 57], such as organophosphates, the pyrethroid cypermethrin, and chlorantraniliprole [64]. *CYP9Q3* is known to metabolize tau-fluvalinate, and *CYP9Q1* and *CYP9Q2* are responsible for degrading bifenthrin [15]. However, the function of *CYP4C3* has not yet been elucidated in honeybees. A previous study used transcriptome sequencing and bioinformatic analysis to compare transcription levels between a susceptible and resistant strain of *Aedes aegypti*; it was reported that the expression of genes such as *CYP4C3* were significantly upregulated in resistant strain, suggesting the existence of a potential relationship between the expression of genes participating in metabolic processes and insecticide resistance [65]. CarEs include a group of enzymes involved in endocrine control, detoxification, and metabolism of nonpolar carboxyl hydrolases [66–68], such as malathion [66], methyl parathion [69], dichlorvos [70], and thiamethoxam [71, 72]. CarE has six isoforms in *A. mellifera* [73]; three isoforms (CarE1, CarE2, and CarE3) are involved in the metabolism of pesticides [74]. For example, CarE1 is directly involved in the detoxification of imidacloprid [71]. In the current study, the expression levels of *CYP4c3* and *CaE-I1* were upregulated by exposure to amitraz, indicating that these genes may be involved in amitraz degradation in honeybees.

Among the identified DEGs, genes encoding the protein Big Brother, fibrillin-2, Ral GTPase-activating protein, and Brachyury protein are vital for the growth and development of insects. An earlier study reported that Big Brother proteins are required during *Drosophila* development [75], and the fibrillin-like protein AD10 was found to affect wing morphogenesis in *Bombyx mori* [76]. The small GTP-binding protein Ral was reported to control the cytoskeletal structure required for cell shape changes during *Drosophila* development [77], and Brachyury is known to regulate gastrulation in *Drosophila* [78]. Our data show that expression of genes encoding these four proteins was upregulated in response to amitraz treatment. A previous study found that acute exposure to amitraz caused cell death in the honeybee larvae midgut [22]. Thus, we believe that in response to amitraz challenge, honeybees could repair the damage to the midgut by inducing the expression levels of the four aforementioned genes.

## Supporting information

**S1 Table. Details of read counts in each library.**
(XLSX)

**S2 Table. Sequencing sequence statistics and quality control.**
(XLSX)

**S3 Table. Abundance distribution of unigenes.**
(XLSX)

**S4 Table. Differentially-expressed genes.**
(XLSX)

**S5 Table. 23 genes with the most significant differential.**
(XLSX)

## Author Contributions

**Conceptualization:** Liang Ye, Peng Liu, Yujie Zhu.

**Data curation:** Liang Ye, Peng Liu, Lai Li.

**Formal analysis:** Liang Ye, Peng Liu.

**Funding acquisition:** Liang Ye, Peng Liu, Anran Wang.

**Investigation:** Liang Ye, Peng Liu, Anran Wang, Lai Li.

**Methodology:** Liang Ye, Peng Liu, Tengfei Shi.

**Project administration:** Liang Ye, Tengfei Shi, Anran Wang.

**Resources:** Liang Ye, Tengfei Shi, Anran Wang, Yujie Zhu, Lai Li.

**Software:** Liang Ye, Tengfei Shi.

**Supervision:** Liang Ye, Tengfei Shi, Lai Li.

**Validation:** Liang Ye, Tengfei Shi, Yujie Zhu.

**Visualization:** Liang Ye, Yujie Zhu.

**Writing – original draft:** Liang Ye.

**Writing – review & editing:** Liang Ye, Peng Liu, Tengfei Shi, Linsheng Yu.

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
