## [Decision Letter · Decision Letter 0]

27 Nov 2019

PONE-D-19-30792

Transcriptomic Analysis to Elucidate the Response of Honeybees (Hymenoptera: Apidae) to Amitraz Treatment

PLOS ONE

Dear Dr Yu,

Thank you for submitting your manuscript to PLOS ONE. After careful consideration, we feel that it has merit but does not fully meet PLOS ONE’s publication criteria as it currently stands. Therefore, we invite you to submit a revised version of the manuscript that addresses the points raised during the review process.

We would appreciate receiving your revised manuscript by Jan 11 2020 11:59PM. To enhance the reproducibility of your results, we recommend that if applicable you deposit your laboratory protocols in protocols.io, where a protocol can be assigned its own identifier (DOI) such that it can be cited independently in the future. For instructions see: http://journals.plos.org/plosone/s/submission-guidelines#loc-laboratory-protocols

We look forward to receiving your revised manuscript.

Kind regards,

Yulin Gao

Academic Editor

PLOS ONE

gaoyulin@caas.cn

Journal Requirements:

2. We note that you are reporting an analysis of a microarray, next-generation sequencing, or deep sequencing data set. PLOS requires that authors comply with field-specific standards for preparation, recording, and deposition of data in repositories appropriate to their field. Please upload these data to a stable, public repository (such as ArrayExpress, Gene Expression Omnibus (GEO), DNA Data Bank of Japan (DDBJ), NCBI GenBank, NCBI Sequence Read Archive, or EMBL Nucleotide Sequence Database (ENA)). In your revised cover letter, please provide the relevant accession numbers that may be used to access these data. For a full list of recommended repositories, see http://journals.plos.org/plosone/s/data-availability#loc-omics or http://journals.plos.org/plosone/s/data-availability#loc-sequencing.

 Review Comments to the Author

Reviewer #1: Amitraz, an acaricide to control varroa mites, affects learning, memory, immunity, and various other physiological processes to honeybees. In this manuscript, the researchers investigated the transcriptome profiles of honeybees after exposure to amitraz for a certain days. The results of this study is potential for further understanding the molecular mechanisms underlying the function of amitraz on honeybees in the future. This manuscript is acceptable, if the authors correct the following mistakes and anwer the related questions.

Comments to this manuscript:

1. In the part of Amitraz Treatment

(1) the authors did not mention where was the Amitraz used in this experiment from, and lack of the neccessary information of the concentration or the status of purity of the chemical. Please add the related information.

(2) In line 89. The authors conducted the experiment with the concentration of amitraz 9.4 mg/L, the problem is whether in this experiment, 9.4 mg/L of the amitraz is the sublethal doses to honeybees. Enven though one reference (Ref 24) listed in the manuscript has shown that in the previous study 9.4 mg/L of the amitraz is the sublethal doses to honeybees. For the reason that in this study, the amitraz maybe different with the chemical which the reference listed. Therefore, please add related information that amitraz 9.4 mg/L is truly sublethal to honeybees in this experiment.

(3) The authors used expression of the concentration of amitraz both of “sublethal” and “subchronic”. Are there any difference between sublethal and subchronic?

(4) In line 95-96, the authors treated the sample with the method “After 10 d, all bees were collected and placed at 4°C for 5 min to make them dizzy; the bees were then dissected on ice, and the midgut was removed and stored at −80°C”. Why did the authors not use liquid nitrogen to quickly ice the sample before the bees were dissected on ice and stored at -80°C?

2. Line 155-157. “mapped reads were 27,182,392, 21,407,197, 24,815,509, 156 26,929,339, 28,752,629, and 157 27,107,676, respectively”. The number here are confusing, please correct them clearly.

3. Authors choosed midgut and sequenced the midgut by transcriptomic methods to find the changes of the gene expression after exposed the honeybees in amitraz . If possible, please privide reasons why the authors choose the midgut as the target tissue. Some genes such as Antimicrobial peptides (AMPs) normally expressed higher in brain and hymolymph involved in the bees. Please disscussed them accordingly.

4. Please provide the BUSCO analysis of the transcriptome to measure genome assembly and annotation completeness. Moreover, please provide the essential information of the Q20 percentage to show the quality of sequencing reads.

Reviewer #2: This paper presents a descriptive study of transcriptomic analysis of honey bees exposed to amitraz treatment, and there are some merits in doing midgut transcriptome of honey bees exposed to amitraz. However, I have some points that require clarification or rewriting, and I hope these comments may be helpful for authors to improver this manuscript.

Comments

1. In the introduction part, the authors should provide more information about effects of amitraz on honey bees and more information regarding the function of midgut in honey bees.

2. RNA sequencing data must be deposited in NCBI Sequence Read Archive (SRA), and accession numbers must be provided in the paper.

3. Lines 33-34, please specify upregulated or downregulated in honey bees exposed to amitraz?

4. Line 64, the authors used midgut for transcriptome study, here is an example of a section of the intro that is relevant for the study. The authors should have expanded this part more.

5. Line 80, “a health colony”, how to define a colony is health or not? By the number of varroa mites in the colony or by virus titer of honey bees? Please specify more clearly.

6. Line 85, bee bread or pollen? Bee bread means that the authors collected it directly from the comb.

7. Line 94, why 10 days of treatment?

8. Line 114, it seems that primer pairs in table 1 were used for validation of RNA-seq data, why authors list primer information here?

9. Line 126, why the authors didn’t use the latest version of representative genome of Apis mellifera (assembly Amel_HAv3.1)?

10. Lines 139-140, why these genes were chosen for validating the RNA-seq data?

11. Line, 141, it’s better to use two reference genes for qPCR analysis to obtain rigorous validation data.

12. Line 176, “23 most significantly differentially expressed genes”, these genes were selected by fold change or by p value? Also, the authors said adjusted p value or corrected p value were used for analysis, but in Supp Table, only p values were shown.

13. Line 199, did the authors do amplification efficiencies analysis before using the 2−ΔΔCt method?

14. Lines 226-236, this part of discussion is poor, and the authors just listed references, but didn’t discuss their data with previous studies well.

15. Line 252, “46 CYPs…”, again I suggest the authors should check the latest version of representative genome of Apis mellifera.

16. Overall, in the discussion part, the authors should discuss more about the biological significances of their data in honey bees. Also, the authors discuss little about GO function and KEGG Pathway analysis.

17. Figure 1 is the same to figure 2? Please check.

---

## [Author Response · Author response to Decision Letter 0]

11 Dec 2019

Response to PLOS ONE

Q: 1. We note the following previous request has not been addressed:

* We note that you are reporting an analysis of a microarray, next-generation sequencing, or deep sequencing data set. PLOS requires that authors comply with field-specific standards for preparation, recording, and deposition of data in repositories appropriate to their field. Please upload these data to a stable, public repository (such as ArrayExpress, Gene Expression Omnibus (GEO), DNA Data Bank of Japan (DDBJ), NCBI GenBank, NCBI Sequence Read Archive, or EMBL Nucleotide Sequence Database (ENA)). In your revised cover letter, please provide the relevant accession numbers that may be used to access these data.

A: Thank you for great suggestions. The sequencing data are available in the SRA database (SRR10595519-SRR10595524) of the NCBI system.

Response to Reviewers #1

1. In the part of Amitraz Treatment

Q: (1) the authors did not mention where was the Amitraz used in this experiment from, and lack of the neccessary information of the concentration or the status of purity of the chemical. Please add the related information.

A: Thank you for your suggests. We had added relevant information in the article.

Q: (2) In line 89. The authors conducted the experiment with the concentration of amitraz 9.4 mg/L, the problem is whether in this experiment, 9.4 mg/L of the amitraz is the sublethal doses to honeybees. Enven though one reference (Ref 24) listed in the manuscript has shown that in the previous study 9.4 mg/L of the amitraz is the sublethal doses to honeybees. For the reason that in this study, the amitraz maybe different with the chemical which the reference listed. Therefore, please add related information that amitraz 9.4 mg/L is truly sublethal to honeybees in this experiment.

A: It is a great suggests. Thank you! We had added the survival curve for treatment for the article (Fig. 1). Analysis this tab, we found that 9.4 mg/L of amitraz was truly sublethal to honeybees in this experiment.

Q: (3) The authors used expression of the concentration of amitraz both of “sublethal” and “subchronic”. Are there any difference between sublethal and subchronic?

A: Good suggest. Sublethal and subchronic are difference. Sublethal means having an effect less than lethal and is divided into acute and chronic. Subchronic means chronic sublethal. In article, the means of sublethal and subchronic was coinciding with reference.

Q: (4) In line 95-96, the authors treated the sample with the method “After 10 d, all bees were collected and placed at 4°C for 5 min to make them dizzy; the bees were then dissected on ice, and the midgut was removed and stored at −80°C”. Why did the authors not use liquid nitrogen to quickly ice the sample before the bees were dissected on ice and stored at -80°C?

A: Thank you for your suggests. We had used liquid nitrogen to quickly freeze the sample before the bees were dissected on ice and stored at -80°C, but not write in the article before and the detail information had been provided in materials and methods.

Q: 2. Line 155-157. “mapped reads were 27,182,392, 21,407,197, 24,815,509, 156 26,929,339, 28,752,629, and 157 27,107,676, respectively”. The number here are confusing, please correct them clearly.

A: Good suggestion. Thank you, This error has been corrected in article.

Q: 3. Authors choosed midgut and sequenced the midgut by transcriptomic methods to find the changes of the gene expression after exposed the honeybees in amitraz. If possible, please privide reasons why the authors choose the midgut as the target tissue. Some genes such as Antimicrobial peptides (AMPs) normally expressed higher in brain and hymolymph involved in the bees. Please disscussed them accordingly.

A: Midgut is an absorptive organ and the first affected organ for chemical compounds, additionally, the midgut epithelium is responsible for detoxification of ingested xenobiotics. We choose the midgut as the target tissue. AMPs genes normally expressed higher in brain and hymolymph involved in the honey bee, but in this study two AMP genes, encoding apidaecin (Apid1) and glycine-rich cell wall structural protein, were induced in amitraz-treated bees. The gene expression levels of AMPs will affect immunocompetence, therefore in our article we discussed the changes of AMPs gene expression in midgut.

Q: 4. Please provide the BUSCO analysis of the transcriptome to measure genome assembly and annotation completeness. Moreover, please provide the essential information of the Q20 percentage to show the quality of sequencing reads.

A: Thank you for great suggestions. All necessary informations about the Q20 percentage are provided in S2_Table. 

Response to Reviewers #2

Q: 1. In the introduction part, the authors should provide more information about effects of amitraz on honey bees and more information regarding the function of midgut in honey bees. .

A: Great suggestions. Some detail information of the effects of amitraz and function of midgut on honeybees are provided in introduction.

Q: 2. RNA sequencing data must be deposited in NCBI Sequence Read Archive (SRA), and accession numbers must be provided in the paper.

A: Thank you for great suggestions. The sequencing data are available in the SRA database (SRR10595519-SRR10595524) of the NCBI system.

Q: 3. Lines 33-34, please specify upregulated or downregulated in honey bees exposed to amitraz?

A: Thank you for great suggestions. The specify DEGs are showed in Table S4 in results. 

Q: 4. Line 64, the authors used midgut for transcriptome study, here is an example of a section of the intro that is relevant for the study. The authors should have expanded this part more.

A: We have expanded this part more.

Q: 5. Line 80, “a health colony”, how to define a colony is health or not? By the number of varroa mites in the colony or by virus titer of honey bees? Please specify more clearly.

A: Greet suggests. All honeybees were collected from a strong colony. All bee samples were strong and not infected with the parasite. 

Q: 6. Line 85, bee bread or pollen? Bee bread means that the authors collected it directly from the comb.

A: We provided pollen to feed honeybees in this study and revised it inmethod part.

Q: 7. Line 94, why 10 days of treatment?

A: We are according to methods previously reported by Shi et al.(2017), and many studies research the effect of xenobiotics on honey bee by treated 10 days(O’Neal, Scott T et al., 2017; Du Yali et al., 2019) 

Q: 8. Line 114, it seems that primer pairs in table 1 were used for validation of RNA-seq data, why authors list primer information here?

A: Sorry. Table 1 was appeared in the wrong place and we have revised it in article.

Q: 9. Line 126, why the authors didn’t use the latest version of representative genome of Apis mellifera (assembly Amel_HAv3.1)?

A: Sorry. Actually we truly use the latest version of representative genome of Apis mellifera (assembly Amel_HAv3.1) and we have revised it in article

Q: 10. Lines 139-140, why these genes were chosen for validating the RNA-seq data?

A: Firstly, the expression of these genes were significant different. Secondly, they were involved in the immunity and growth of honeybee.

Q: 11. Line, 141, it’s better to use two reference genes for qPCR analysis to obtain rigorous validation data.

A: Thank you for suggests. We had added an experiment by used other reference gene (β-actin) for qPCR analysis and showed in results.

Q: 12. Line 176, “23 most significantly differentially expressed genes”, these genes were selected by fold change or by p value? Also, the authors said adjusted p value or corrected p value were used for analysis, but in Supp Table, only p values were shown.

A: Thank you for suggests. Yes, these genes were selected by p value. We had added information and improved our Supp Table 5. 

Q: 13. Line 199, did the authors do amplification efficiencies analysis before using the 2−ΔΔCt method?

A: Yes, we did. Thank you.

Q: 14. Lines 226-236, this part of discussion is poor, and the authors just listed references, but didn’t discuss their data with previous studies well.

A: Thank you for great suggestions. The discussion had been enriched.

Q: 15. Line 252, “46 CYPs…”, again I suggest the authors should check the latest version of representative genome of Apis mellifera.

A: Thank you for great suggestions. We check the number, there are only 46 CYPs in genome of A. mellifera (Johnson 2012), and we do not found new CYP genes in honeybee at 2012 to 2019. However we revised the sentence to “Honey bee are known to have the far fewer number of CYP family genes than other insect.”

Q: 16. Overall, in the discussion part, the authors should discuss more about the biological significances of their data in honey bees. Also, the authors discuss little about GO function and KEGG Pathway analysis.

A: Great suggestions. The discussion part had been enriched

Q: 17. Figure 1 is the same to figure 2? Please check.

A: Sorry. The fig 1 was the same to fig 2, we deleted the fig 1 and replaced it with venn diagram.

---

## [Decision Letter · Decision Letter 1]

2 Jan 2020

PONE-D-19-30792R1

Transcriptomic Analysis to Elucidate the Response of Honeybees (Hymenoptera: Apidae) to Amitraz Treatment

PLOS ONE

Dear Dr Yu,

Thank you for submitting your manuscript to PLOS ONE. After careful consideration, we feel that it has merit but does not fully meet PLOS ONE’s publication criteria as it currently stands. Therefore, we invite you to submit a revised version of the manuscript that addresses the points raised during the review process.

We would appreciate receiving your revised manuscript by Feb 16 2020 11:59PM. To enhance the reproducibility of your results, we recommend that if applicable you deposit your laboratory protocols in protocols.io, where a protocol can be assigned its own identifier (DOI) such that it can be cited independently in the future. For instructions see: http://journals.plos.org/plosone/s/submission-guidelines#loc-laboratory-protocols

We look forward to receiving your revised manuscript.

Kind regards,

Yulin Geo

Academic Editor

PLOS ONE

Reviewers' comments:

Reviewer's Responses to Questions

**Comments to the Author**

1. If the authors have adequately addressed your comments raised in a previous round of review and you feel that this manuscript is now acceptable for publication, you may indicate that here to bypass the “Comments to the Author” section, enter your conflict of interest statement in the “Confidential to Editor” section, and submit your "Accept" recommendation.

Reviewer #1: All comments have been addressed

Reviewer #2: (No Response)

2. Is the manuscript technically sound, and do the data support the conclusions?

Reviewer #1: Yes

Reviewer #2: Partly

3. Has the statistical analysis been performed appropriately and rigorously? 

Reviewer #1: Yes

Reviewer #2: I Don't Know

4. Have the authors made all data underlying the findings in their manuscript fully available?

Reviewer #1: Yes

Reviewer #2: Yes

5. Is the manuscript presented in an intelligible fashion and written in standard English?

Reviewer #1: Yes

Reviewer #2: Yes

6. Review Comments to the Author

Reviewer #1: The authors corrected all the mistakes in the manuscript according to the reviewers' comments. It is acceptable for publication.

Reviewer #2: Comments for PONE-D-19-30792R1:

1. Page 5, lines 87-88, the first major methodological concern that I have concerns the use of just one colony for the study. I see this as a major flaw in the experimental design, as everybody working with social insects like honeybees should know that some replications is always needed at the colony level in addition to replication at the individual level. At least two or three colonies are always recommended for any type of study on social insects if the authors want to obtain results that can be generalized. In fact, all individuals within the same colonies are highly related genetically, and therefore there might be a specific effect of that colony affecting the results and the authors will not be able to identify these effects.

2. Page 5, line 102, a second major methodological concern that I have concerns the use of acetone in amitraz-treated bees, but only sucrose–water solution was used in untreated bees. Amitraz was dissolved in acetone, so acetone-treated control bees should be used in the study. Therefore, acetone might have effects on gene expression of honey bees, and the authors will not be able to distinguish whether changing in gene expression is due to amitraz or acetone.

3. Page 8, line 159, because 2−ΔΔCt method was used, please show amplification efficiency of each primer in the manuscript.

4. Page 13, line 240-241, only RPS5 was used?

7. PLOS authors have the option to publish the peer review history of their article (what does this mean?). If published, this will include your full peer review and any attached files.

Reviewer #1: No

Reviewer #2: No

---

## [Author Response · Author response to Decision Letter 1]

2 Jan 2020

Response to Reviewers #1

Thank you for your comments.

Response to Reviewers #2

Q: 1. Page 5, lines 87-88, the first major methodological concern that I have concerns the use of just one colony for the study. I see this as a major flaw in the experimental design, as everybody working with social insects like honeybees should know that some replications is always needed at the colony level in addition to replication at the individual level. At least two or three colonies are always recommended for any type of study on social insects if the authors want to obtain results that can be generalized. In fact, all individuals within the same colonies are highly related genetically, and therefore there might be a specific effect of that colony affecting the results and the authors will not be able to identify these effects.

A: Great suggestions. Compared to treatment in only one colony, several colonies have a advantage of genetical diversity and avoid a specific effect of one colony. Some study researched the influence of environment stress above physiological response[1-3]. However, in the past decade several studies choose honey bees which was from same colony as samples [4-8]. Reference those studies, we explored the effect of amitraz on honeybees to by treat the honey bees from same colony.

Q: 2. Page 5, line 102, a second major methodological concern that I have concerns the use of acetone in amitraz-treated bees, but only sucrose–water solution was used in untreated bees. Amitraz was dissolved in acetone, so acetone-treated control bees should be used in the study. Therefore, acetone might have effects on gene expression of honey bees, and the authors will not be able to distinguish whether changing in gene expression is due to amitraz or acetone.

A: Thank you for great suggestions. We had acetone-treated control bees in the study, and revised in materials and methods.

Q: 3. Page 8, line 159, because 2−ΔΔCt method was used, please show amplification efficiency of each primer in the manuscript.

A: Thank you for great suggestions. We added amplification efficiency of each primer in Table 1

Q: 4. Page 13, line 240-241, only RPS5 was used?

A: Sorry, this a mistake here. We truly use two reference genes (Rps5 and β-actin) for qPCR and we have revised it in article.

1. Rutter L, Carrillo-Tripp J, Bonning BC, Cook D, Toth AL, Dolezal AG. Transcriptomic responses to diet quality and viral infection in Apis mellifera. BMC Genomics. 2019;20(1):412. Epub 2019/05/24. doi: 10.1186/s12864-019-5767-1. PubMed PMID: 31117959; PubMed Central PMCID: PMCPMC6532243.

2. Zhu L, Qi S, Xue X, Niu X, Wu L. Nitenpyram disturbs gut microbiota and influences metabolic homeostasis and immunity in honey bee (Apis mellifera L.). Environ Pollut. 2019;258:113671. Epub 2019/12/20. doi: 10.1016/j.envpol.2019.113671. PubMed PMID: 31855676.

3. Mondet F, Rau A, Klopp C, Rohmer M, Severac D, Le Conte Y, et al. Transcriptome profiling of the honeybee parasite Varroa destructor provides new biological insights into the mite adult life cycle. BMC Genomics. 2018;19(1):328. Epub 2018/05/08. doi: 10.1186/s12864-018-4668-z. PubMed PMID: 29728057; PubMed Central PMCID: PMCPMC5936029.

4. Alburaki M, Karim S, Lamour K, Adamczyk J, Stewart SD. RNA-seq reveals disruption of gene regulation when honey bees are caged and deprived of hive conditions. The Journal of Experimental Biology. 2019;222(18). doi: 10.1242/jeb.207761.

5. Christen V, Schirrmann M, Frey JE, Fent K. Global Transcriptomic Effects of Environmentally Relevant Concentrations of the Neonicotinoids Clothianidin, Imidacloprid, and Thiamethoxam in the Brain of Honey Bees ( Apis mellifera). Environ Sci Technol. 2018;52(13):7534-44. Epub 2018/06/01. doi: 10.1021/acs.est.8b01801. PubMed PMID: 29851480.

6. Shi TF, Wang YF, Liu F, Qi L, Yu LS. Sublethal Effects of the Neonicotinoid Insecticide Thiamethoxam on the Transcriptome of the Honey Bees (Hymenoptera: Apidae). J Econ Entomol. 2017;110(6):2283-9. Epub 2017/10/19. doi: 10.1093/jee/tox262. PubMed PMID: 29040619.

7. Emery O, Schmidt K, Engel P. Immune system stimulation by the gut symbiont Frischella perrara in the honey bee (Apis mellifera). Mol Ecol. 2017;26(9):2576-90. Epub 2017/02/17. doi: 10.1111/mec.14058. PubMed PMID: 28207182.

8. Mao W, Schuler MA, Berenbaum MR. Disruption of quercetin metabolism by fungicide affects energy production in honey bees (Apis mellifera). Proc Natl Acad Sci U S A. 2017;114(10):2538-43. Epub 2017/02/15. doi: 10.1073/pnas.1614864114. PubMed PMID: 28193870; PubMed Central PMCID: PMCPMC5347564.

---

## [Decision Letter · Decision Letter 2]

17 Jan 2020

PONE-D-19-30792R2

Transcriptomic Analysis to Elucidate the Response of Honeybees (Hymenoptera: Apidae) to Amitraz Treatment

PLOS ONE

Dear Dr Yu,

Thank you for submitting your manuscript to PLOS ONE. After careful consideration, we feel that it has merit but does not fully meet PLOS ONE’s publication criteria as it currently stands. Therefore, we invite you to submit a revised version of the manuscript that addresses the points raised during the review process.

We would appreciate receiving your revised manuscript by Mar 02 2020 11:59PM. To enhance the reproducibility of your results, we recommend that if applicable you deposit your laboratory protocols in protocols.io, where a protocol can be assigned its own identifier (DOI) such that it can be cited independently in the future. For instructions see: http://journals.plos.org/plosone/s/submission-guidelines#loc-laboratory-protocols

We look forward to receiving your revised manuscript.

Kind regards,

Yulin Gao

Academic Editor

PLOS ONE

 Review Comments to the Author

Reviewer #2: Comments for PONE-D-19-30792R2:

1. Lines 87-88, ‘a healthy colony’ should be ‘an apparently healthy colony’.

2. Lines 153, where is the total RNA for qPCR validation from? How many replicates were used for qPCR validation? More details needed here.

3. Line 157, for Table 1, the author should cite references if primer sequences were from published papers. Also, please check the primer sequences carefully, the reverse primer sequences of actin is wrong.

4. Line 181, ‘…were calculated using result in S2 Table’? Please check and rephrase the sentence.

5. Line 224, I got confused with the qPCR analysis result, the authors calculated the expression level of the gene of interest using RPS5 and actin respectively, and the qPCR data was shown in fig5a and 5b respectively. Actually, the authors should calculate the geometric mean of references genes first and then calculate the expression level of each target gene based on the geometric mean of references genes. Please refer to (Garrido et al., 2013; Wu et al., 2017).

6. Lines 260-262, Please cite references here. Also, should be “… in different developmental stages of honey bees”.

7. Lines 262-264, this sentence just repeats the content of lines 260-262 and is redundant. Please delete or rephrase the sentence.

8. Line 265, ‘Thiamethoxam treated honey…exposed to either thiamethoxam…’, again further treated with the same pesticide? Please check.

9. Lines 265-266, gene names and Nosema should be printed in italics.

10. Line 267, why the authors conclude that glycine-rich cell wall structural protein is antimicrobial peptides.

11. Lines 287-290, the content regarding phytochemical metabolism, furanocoumarins, and quercetin is not closely related to the topic of this study. Please delete.

12. Line 297, ‘…were significantly upregulated in susceptible strain…’? Please specify.

13. As I mentioned earlier, the authors should discuss more about GO function and KEGG pathway analysis in the discussion part.

14. The author should incorporate the response to reviewer comments into the revised manuscript. For example, the justification for only one colony was used for sample preparation in the study.

---

## [Author Response · Author response to Decision Letter 2]

19 Jan 2020

Response to Reviewers #2

Q: 1. Lines 87-88, ‘a healthy colony’ should be ‘an apparently healthy colony’.

A: Thank you for great suggestions. We have revised it in article

Q: 2. Lines 153, where is the total RNA for qPCR validation from? How many replicates were used for qPCR validation? More details needed here.

A: Great suggestions. The total RNA for qPCR is the same as the total RNA for RNA-seq. We used three replicates for qPCR validation. We have added relevant information in the article.

Q: 3. Line 157, for Table 1, the author should cite references if primer sequences were from published papers. Also, please check the primer sequences carefully, the reverse primer sequences of actin is wrong.

A: Sorry. The reverse primer sequences of actin is wrong, we have revised it and cite references in article. 

Q: 4. Line 181, ‘…were calculated using result in S2 Table’? Please check and rephrase the sentence.

A: Thank you for good suggestions. We have rephrased this sentence in article.

Q: 5. Line 224, I got confused with the qPCR analysis result, the authors calculated the expression level of the gene of interest using RPS5 and actin respectively, and the qPCR data was shown in fig5a and 5b respectively. Actually, the authors should calculate the geometric mean of references genes first and then calculate the expression level of each target gene based on the geometric mean of references genes. Please refer to (Garrido et al., 2013; Wu et al., 2017).

A: Thank you for great suggestions. Refer to this way, we make a new fig to show our qPCR results. 

Q: 6. Lines 260-262, Please cite references here. Also, should be “… in different developmental stages of honey bees”.

A: Great suggestions. We have revised it and add relevant information in the article.

Q: 7. Lines 262-264, this sentence just repeats the content of lines 260-262 and is redundant. Please delete or rephrase the sentence.

A: It is a great suggests. Thank you! We have rephrased the sentence in the article.

Q: 8. Line 265, ‘Thiamethoxam treated honey…exposed to either thiamethoxam…’, again further treated with the same pesticide? Please check.

A: Thank you for great suggestions. We check this referenc, they again further treated with the same pesticide truly.

Q: 9. Lines 265-266, gene names and Nosema should be printed in italics.

A: Good suggestions. We have revised it in article

Q: 10. Line 267, why the authors conclude that glycine-rich cell wall structural protein is antimicrobial peptides.

A: Sorry, we have revised it in article

Q: 11. Lines 287-290, the content regarding phytochemical metabolism, furanocoumarins, and quercetin is not closely related to the topic of this study. Please delete.

A: Thank you for great suggestions. We have deleted it in article

Q: 12. Line 297, ‘…were significantly upregulated in susceptible strain…’? Please specify.

A: It is a great suggests. Thank you! We have specified it in article.

Q: 13. As I mentioned earlier, the authors should discuss more about GO function and KEGG pathway analysis in the discussion part.

A: Great suggestions. The discussion part had been enriched

Q: 14. The author should incorporate the response to reviewer comments into the revised manuscript. For example, the justification for only one colony was used for sample preparation in the study.

A: Great suggestions. We have incorporated the response to reviewer comments into the revised manuscript.

---

## [Editor Report · Decision Letter 3]

28 Jan 2020

Transcriptomic Analysis to Elucidate the Response of Honeybees (Hymenoptera: Apidae) to Amitraz Treatment

PONE-D-19-30792R3

Dear Dr. Yu,

We are pleased to inform you that your manuscript has been judged scientifically suitable for publication and will be formally accepted for publication once it complies with all outstanding technical requirements.

With kind regards,

Yulin Gao

Academic Editor

PLOS ONE

---

## [Editor Report · Acceptance letter]

28 Feb 2020

PONE-D-19-30792R3 

Transcriptomic analysis to elucidate the response of honeybees (Hymenoptera: Apidae) to amitraz treatment 

Dear Dr. Yu:

I am pleased to inform you that your manuscript has been deemed suitable for publication in PLOS ONE. Congratulations! Your manuscript is now with our production department. 

With kind regards,

on behalf of

Dr. Yulin Gao 

Academic Editor

PLOS ONE